# Application of Photodynamic Therapy with 5-Aminolevulinic Acid to Extracorporeal Photopheresis in the Treatment of Cutaneous T-Cell Lymphoma: A First-in-Human Phase I/II Study [note 1]

**DOI:** 10.3390/pharmaceutics16060815

**Published:** 2024-06-16

**Authors:** Eidi Christensen, Olav Andreas Foss, Toril Holien, Petras Juzenas, Qian Peng

**Affiliations:** 1Department of Dermatology, St. Olav’s Hospital, Trondheim University Hospital, 7030 Trondheim, Norway; 2Department of Clinical and Molecular Medicine, Norwegian University of Science and Technology (NTNU), 7030 Trondheim, Norway; 3Department of Pathology, The Norwegian Radium Hospital, Oslo University Hospital, 0310 Oslo, Norway; petras.juzenas@rr-research.no (P.J.); qian.peng@rr-research.no (Q.P.); 4Department of Orthopaedic Surgery, Clinic of Orthopaedic, Rheumatology and Dermatology, St. Olav’s Hospital, Trondheim University Hospital, 7030 Trondheim, Norway; olav.foss@ntnu.no; 5Department of Biomedical Laboratory Science, Norwegian University of Science and Technology (NTNU), 7030 Trondheim, Norway; toril.holien@ntnu.no; 6Department of Immunology and Transfusion Medicine, St. Olav’s Hospital, Trondheim University Hospital, 7030 Trondheim, Norway; 7Department of Optical Science and Engineering, School of Information Science and Technology, Fudan University, Shanghai 200433, China

**Keywords:** 5-aminolevulinic acid, ALA-based photodynamic therapy, phototherapy, extracorporeal photopheresis, cutaneous T-cell lymphoma

## Abstract

Extracorporeal photopheresis (ECP) is a therapeutic modality used for T-cell-mediated disorders. This approach involves exposing isolated white blood cells to photoactivatable 8-methoxypsoralen (8-MOP) and UVA light, aiming to induce apoptosis in T-cells and thereby modulate immune responses. However, conventional 8-MOP-ECP lacks cell selectivity, killing both healthy and diseased cells, and has shown limited treatment efficacy. An alternative approach under investigation involves the use of 5-aminolevulinic acid (ALA) in conjunction with light, referred to as ALA-based photodynamic therapy. Our previous ex vivo studies suggest that ALA-ECP exhibits greater selectivity and efficiency in killing T-cells derived from patients with T-cell-mediated disorders compared to those treated with 8-MOP-ECP. We have conducted a clinical phase I–(II) study evaluating ALA-ECP safety and tolerability in cutaneous T-cell lymphoma (CTCL). Here, 20 ALA-ECP treatments were administered to one CTCL patient, revealing no significant changes in vital signs. Two adverse events were reported; both evaluated by the Internal Safety Review Committee as non-serious. In addition, five conceivable events with mainly mild symptoms took place. During the study period, a 53% reduction in skin involvement and a 50% reduction in pruritus was observed. In conclusion, the results indicate that ALA-ECP treatment is safe and well tolerated.

## 1. Introduction

Extracorporeal photopheresis (ECP) is an immune-modulating therapy primarily targeting the T-cells of the immune system. It was approved more than three decades ago as a standard therapy for patients with advanced refractory cutaneous T-cell lymphoma (CTCL) and has gradually extended to the treatment of patients with other T-cell-mediated diseases [1,2].

CTCL constitutes a heterogeneous group of lymphoproliferative disorders originating from malignant T-lymphocytes. Predominant variants include mycosis fungoides (MF) and its leukemic variant, Sézary Syndrome (SS) [3]. MF manifests as skin patches, plaques, and tumors, while SS is characterized by erythematous skin involvement and the presence of malignant T-cells in peripheral blood. The initial stages of MF typically respond to topical or phototherapy interventions. In progressive cases, systemic treatments such as interferon (IFN)-α and bexarotene are commonly employed. ECP can serve as a beneficial monotherapy or adjunct to immunotherapy in managing erythrodermic MF and SS. 

ECP involves leukapheresis, during which leukocytes are separated from the whole blood [2]. The isolated fraction of white blood cells (buffy coat) is then exposed to the photosensitizing agent 8-methoxypsoralen (8-MOP) and ultraviolet A (UVA) light before being reintroduced into the patient. This process induces DNA crosslinking within the treated cells, ultimately leading to apoptotic cell death [2]. Additional mechanisms of action include transimmunization, modulation of cytokine profiles, and shifts in immune responses [4]. However, the non-selective binding of 8-MOP to both diseased and normal cells makes the current ECP treatment disadvantageous, as it results in apoptosis of both types of cells upon UVA irradiation. Moreover, 8-MOP-ECP has shown long treatment duration, high costs, and only partial efficacy in many patients, which in turn calls for the exploration of alternative photosensitizers with greater selectivity and efficiency when used for ECP.

5-Aminolevulinic acid (ALA), a naturally occurring amino acid and heme precursor, is metabolized intracellularly to endogenous porphyrin photosensitizers, predominantly protoporphyrin IX (PpIX) [5]. This process results in selective accumulation of PpIX in proliferative tumor cells as well as activated immune cells mainly due to their enzymatic alterations in the heme biosynthesis. Upon photoactivation of PpIX in the presence of oxygen, reactive oxygen species are formed, inducing apoptosis and necrosis in the targeted cells [6,7].

ALA-mediated photodynamic therapy (PDT) is established for topical treatment of non-melanoma skin cancer, and oral ALA is also used clinically for glioma photodetection and Barrett’s esophagus treatment [5,8,9,10]. Moreover, ALA plus UVA has been shown to be more effective than 8-MOP with UVA in killing activated T-cells from patients with CTCL and chronic graft-versus-host disease (cGvHD) [11]. Thus, modifying standard ECP with ALA could improve treatment efficacy, potentially reducing treatment frequency and duration.

The primary aim of this study was to investigate the safety and tolerability of ALA-ECP treatment in CTCL. In addition, scoring of various disease-related symptoms was carried out during the study period. 

## 2. Material and Methods

### 2.1. Design

We have conducted a phase I–(II) study to evaluate the safety and tolerability of ALA used for ECP among patients diagnosed with T-cell-mediated disorders [12]. This prospective, open-label, single-center phase I-(II) study received approval from the Regional Committee for Medical Research Ethics (REK-Nord 2014/2316) and the Norwegian Medicines Agency (14/16760-29), the national regulatory authority. It was registered at www.clinicaltrials.gov under the identifier NCT03109353. The Clinical Research Unit Central Norway, Norwegian University of Science and Technology (NTNU) provided study monitoring.

The previously published paper on ALA-ECP in patients with cGvHD offers a comprehensive description of the methods used [12]. The present report focuses on the findings related to the use of ALA-ECP in CTCL.

### 2.2. Inclusion and Exclusion 

Patients with CTCL undergoing treatment at St. Olavs Hospital, Trondheim University Hospital, with 8-MOP-ECP for a minimum of 3 months and exhibiting inadequate treatment response, were considered eligible. The inadequate response was defined as either progressive disease (disease worsening from baseline skin symptoms, blood, or lymph nodes scores after 3–6 months), stable disease (lack of response after 3–6 months), or minimal response (less than 50% reduction in skin scores and/or CD4/CD8 ratio, or disappearance of peripheral blood clone after 3–6 months).

Exclusion criteria encompassed individuals under 18 years, weighing less than 40 kg, with photosensitive conditions like porphyria or known hypersensitivity to 5-ALA or porphyrins, aphakia, pregnancy or lactation, polyneuropathy, cardiac or pulmonary disorders, uncontrolled infection or fever, history of heparin-induced thrombocytopenia, absolute neutrophil count < 1 × 10^9^/L, platelet count < 20 × 10^9^/L, elevated aspartate transaminase (AST), alanine transaminase (ALT), bilirubin, or prothrombin time international normalized ratio (PT-INR) ≥ 3× upper limit of normal, significant electrocardiogram (ECG) findings, inability to adhere to study procedures, or conditions influencing safety or participation.

Patients were eligible for up to 20 ALA-ECP treatments, equating to 10 treatment cycles within one year, with each cycle comprising two consecutive-day treatments. Treatment scheduling was flexible based on patient symptoms and response.

### 2.3. Procedures

The CELLEX^®^ machine (Therakos, Mallinckrodt Pharmaceuticals, Raritan, NL, USA) was used to perform ECP in this study. Whole blood was collected via venous access. Leukocyte-enriched blood fraction (buffy) was separated by centrifugation and collected into a treatment bag (buffy bag), while other blood components were returned to the patient. Commercially available ALA-hydrochloride powder (Gliolan^®^ photonamic GmbH & Co. KG, Pinneberg, Germany) was reconstituted by the Pharmacy at St. Olavs Hospital to a 30 mg/mL solution in 0.9% NaCl immediately before each treatment. The volume of ALA-HCL added to the buffy coat was adjusted based on the collected buffy volume to achieve a treatment dose of 0.168% w/v (10 mM). After adding the ALA solution to the buffy bag, a 1-h incubation period allowed for intracellular PpIX production before UVA light exposure. The bag was placed on a rocker device during incubation to prevent cell clotting. The CELLEX^®^ machine system calculated and set the UVA light exposure time based on factors including buffy hematocrit percentage, treatment volume, and remaining lamp life. Subsequently, the treated buffy was reinfused into the patient.

Buffy coat samples were collected to investigate ALA-induced PpIX levels after incubation with ALA for one hour at room temperature [13,14]. Plasma samples for PpIX determination were also obtained immediately after reinfusion of the ex vivo ALA-ECP-treated buffy coat and again 24 h post-treatment. PpIX fluorescence was measured using a luminescence spectrometer (LS50B, Perkin-Elmer, Norwalk, CT, USA). The sensitivity of the measurement allowed the detection of PpIX at a concentration as low as approximately 3 nM.

### 2.4. Safety and Tolerability

Safety monitoring involved vital signs (blood pressure, pulse, temple temperature) and laboratory analyses of blood and urine samples obtained at each treatment session. Flow cytometry was used to determine the percentage of CD4^+^ and CD8^+^ cells. ECG monitoring and reporting of adverse events and evaluation of effect took place every 3 months. 

Laboratory analyses included hematology and clinical chemistry parameters and were performed at the Department of Medical Biochemistry, St. Olavs Hospital, with their reference values. Additional blood samples for AST, ALT, bilirubin, and INR were collected after each treatment cycle since transient liver effects are commonly associated with oral ALA administration. Urine was analyzed at the Department of Dermatology using dipstick analysis.

An Internal Safety Review Committee (ISRC) comprising three members was appointed to assess safety and tolerability data from blood samples and evaluate reported adverse events (AEs). Conceivable adverse events associated with systemic ALA treatment included nausea, vomiting, headache, photosensitivity, and chills. Patients were provided with diaries to record adverse events occurring between visits. The severity of all events was graded according to the Common Terminology Criteria for Adverse Events (CTCAE) v4.03. 

### 2.5. Response Assessments

Response assessments were conducted approximately every 3 months following the initial ALA-ECP treatment, with the baseline conditions serving as a reference. Evaluations by investigators and patient-reported outcome measures were employed. The use of immunosuppressive therapy was monitored. Skin assessments were conducted by the investigator utilizing the modified severity-weighted assessment tool (mSWAT) to evaluate skin involvement [15]. mSWAT involves assessing the body surface area (BSA) of each type of MF/SS lesion in 12 body areas, multiplying the BSA of each lesion type by a weighting factor (patch = 1, plaque = 2, and tumor = 3 or 4), and summing the subtotals of each lesion subtype. Pruritus severity was assessed using a visual analog scale (VAS). Patients completed the Skindex-29 and European Organisation for the Research and Treatment of Cancer Quality of Life Questionnaire Core 30 (EORTC30) questionnaires, which comprehensively measure health-related quality of life in patients with skin disease and cancer, respectively [16,17].

### 2.6. Statistical Methods

Descriptive statistics are reported as numbers, percentages, means (min.–max.), and standard deviation (±). Statistical analyses were conducted using IBM SPSS software version 23.

## 3. Results 

### 3.1. Patient and Treatments

Two patients were assessed for availability. One showed rapid progression of disease that required intensive cytostatic treatment in the cancer department. Thus, this patient was excluded from inclusion due to the probable inability to comply with the study procedures. The second patient, a female aged 75 years, diagnosed with CTCL of the SS stage IIIB variant, was enrolled in the study. Prior to study inclusion, the patient had undergone 25 cycles of 8-MOP-ECP treatment. During the study, the patient received methotrexate for CTCL in addition to ECP and topical corticosteroids. 

The patient underwent ALA-ECP treatments at 4-week intervals, receiving a total of 20 treatments. The mean volume of the buffy coat was 141 mL (100–274), and the mean illumination time during ECP was 18 min (9–39).

### 3.2. Safety and Tolerability 

We observed no clinically significant changes in systolic- and diastolic blood pressure, heart rate, or temple temperature measurements during the study period (Table 1).

No clinically abnormal findings were found on physical examination, including palpable lymph nodes larger than 15 mm. Although the patient’s ECG screening showed a right bundle branch block, the cardiologist’s investigation concluded that the finding was not significant, and the ECG recordings remained unchanged during the study period.

Generally, only small variations in laboratory values were observed (Table 2). There was no observation of liver toxicity. 

Analyses of blood samples for AST, ALT, and bilirubin taken before and after each treatment cycle only showed slight variations. There was a modest increase in the PT-INR value on day two of all treatments. More details are presented in Table 1.

PpIX levels were evaluated in 24 buffy coats and 44 plasma samples. Prior to ALA addition, baseline PpIX levels in all buffy coat samples were undetectable. Following 1-h ALA incubation, the PpIX concentration in buffy coat samples was approximately 109 ± 80 nM. After irradiation, the PpIX concentration in these samples was about 165 ± 141 nM. No PpIX was detected in plasma samples collected from the peripheral blood of patients before or after treatment.

The CD4/CD8 ratio decreased from 15.9 at baseline to 7.4 (53%) at the last control.

Urine dipstick tests were negative except on one occasion where it showed plus one on leukocytes. The methotrexate dose was reduced from 20 mg/week to 17.5 mg/week. The patient received treatment with Gabapentin (Neurontin^®^) capsules for nocturnal pruritus from about the sixth treatment cycle and was discontinued after 3 months due to a reduction in pruritus severity.

Together, six conceivable adverse events and two adverse events were reported with either grade 1 or grade 2 severity. ISRC evaluated the two AEs to be non-serious and possibly related to the study medication. More detailed information on AEs is given in Table 3. No grade 3–5 events including any serious adverse event (SAE) were recorded. 

### 3.3. Response Assessments 

Assessment scores collected at the baseline and last control are shown in Table 4. The patient presented with multiple MF skin patches and plaques at baseline but mainly with patches and post-inflammatory hyperpigmentation at the last control. Clinical pictures taken before and after the last treatment are presented in Figure 1, Figure 2, Figure 3 and Figure 4. There was an improvement in the skin score with 69% and in the pruritus score with 50%. Results of quality of life measures showed unaltered or a tendency of improvement in the scores. 

## 4. Discussion

This report presents the outcomes of 20 ALA-ECP treatments in a CTCL patient who was considered to respond inadequately to the standard 8-MOP-ECP. Throughout the duration of the study, no clinically significant alterations in vital signs were noted, and there were no persistent deviations observed in the laboratory parameters. Although common side effects observed from the use of oral ALA are transient rise in some liver enzymes and bilirubin [18,19], this study showed no signs of liver toxicity. The observations of a modest increase in the INR value on day two of treatment might be associated with the blood heparinization during ECP. 

Conventional ECP using 8-MOP is widely recognized for its safety profile with few side effects [2]. Systemic administration of ALA may induce sporadic and mild side effects, such as nausea, fever, or headache [20]. AEs observed during ALA-ECP treatment of the CTCL patient included skin soreness and clotting in the buffy coat. The patient reported skin soreness following the initial treatment cycle, lasting approximately thirteen days, along with transient headache, photosensitivity, and chills. Two further events of chills were also reported on later occasions. The causal relationship between these events and ALA-ECP remains unclear, particularly as they did not consistently manifest across treatments. Although photosensitivity is a recognized side effect of systemic ALA, only once was it reported by the patient after a total of 20 treatments in this study (Table 3). Furthermore, analysis of buffy coat samples detected no PpIX in the plasma collected from the peripheral blood before or after treatment. 

During the initial day of the second cycle of ECP, the occurrence of clotting within the photochamber of the ECP machine was observed. Clotting within the buffy coat is a recognized phenomenon in conventional 8-MOP-ECP and has been previously documented in patients with cGvHD undergoing ALA-ECP [12,21]. The specific contribution of ALA and/or its 1-h incubation period preceding the photoactivation of the buffy coat to abnormal clotting remains uncertain, given the recognition of clotting also in conventional 8-MOP-ECP. The management of clotting during 8-MOP-ECP varies among hospitals, often involving adjustments in anticoagulant therapy, such as heparin, tailored to individual patient needs. In our hospital, clotting events during 8-MOP-ECP are typically addressed by increasing the heparin dosage. Following dosage adjustments, no further instances of clotting were observed in this study. Nevertheless, the surveillance of unforeseen events remains imperative in future larger-scale clinical investigations of ALA-ECP. 

In MF/SS, the pathologic CD4+ cells (helper T cells) undergo clonal expansion, potentially leading to an abnormally elevated CD4/CD8 T-cell ratio in the peripheral blood, and a ratio less than 10 predicts a better response to treatment [22,23]. During the study period, the CD4/CD8 ratio was reduced to 7.4 after the treatment from 15.9 before the treatment. Furthermore, a reduction of the patient’s cutaneous tumor patches and plagues in terms of number and size was documented during the course of the study. These promising therapeutic data may result not only from the selective and efficient direct killing effect on those malignant CD4 cells but also from the induction of systemic anti-tumor immunity.

Apart from allogeneic transplantation, curative therapies for CTCL are lacking [2]. Typically, treatment is focused on palliation and the induction of sustained remission. The primary objective is to diminish or eradicate skin lesions and alleviate pruritus, thus offering symptom relief and enhancing the overall quality of life for the patient. Hence, although safety was the primary focus of this study, skin, pruritus, and quality of life assessments were regularly scored to reflect changes that might occur during the study period. The improvement in these scores reflecting an improvement in the patient’s skin condition appears promising. However, this may not be attributed to ALA-ECP treatment alone. Factors such as the fluctuating course of the disease, the comedication of the patient, and the lack of investigator-blinded controls may have influenced the results.

The rationales for attempting clinical ALA use in ECP of CTCL include (1) Treatment selectivity since significantly more PpIX production occurs in the transformed and activated T-cells compared to normal resting cells [11]. Consequently, upon light irradiation, these abnormally proliferative cells are selectively destroyed while resting normal T-cells remain unaffected so that ALA-ECP may result in more effective elimination of diseased T-cells compared to conventional ECP with 8-MOP/UVA. (2) Induction of anti-tumor immunity [24,25]. ECP with ALA has the potential to stimulate anti-tumor immune responses, contributing to the overall therapeutic efficacy. 

## 5. Conclusions 

The findings of this study indicate that ALA-ECP is safe and well-tolerated, with predominantly mild to moderate side effects occurring infrequently. Such results, in conjunction with prior ALA-ECP outcomes in cGvHD treatment, contribute to expanding our knowledge of ALA-ECP potential in managing T-cell mediated diseases. These findings provide a basis for designing future clinical studies aimed at investigating the safety and efficacy of ALA-ECP and optimizing its treatment protocol in diverse patient populations with similar conditions.

## Figures and Tables

**Figure 1 pharmaceutics-16-00815-f001:**
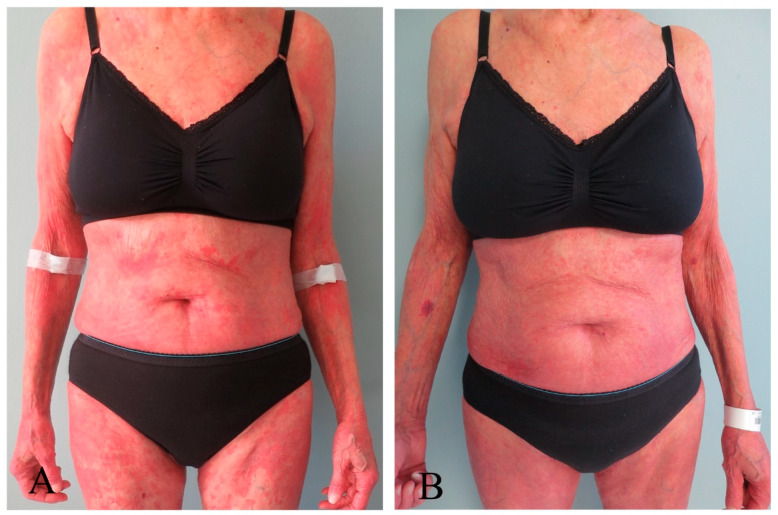
Images of the anterior trunk taken (**A**) before and (**B**) after 20 ALA-ECP treatments, depicting multiple areas with erythema, patches, and plaques before treatment, and paler skin indicative of clinical remission on the upper chest and upper arms, along with predominantly erythematous patches on the abdomen.

**Figure 2 pharmaceutics-16-00815-f002:**
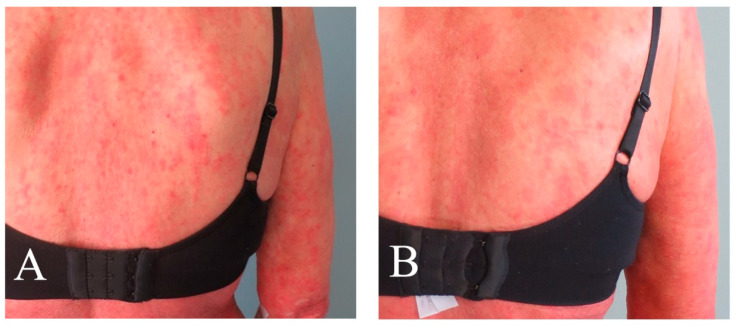
Images with extracts from the patient’s back taken (**A**) before and (**B**) after 20 ALA-ECP treatments, depicting patches and plaques before treatment and predominantly patches after treatment.

**Figure 3 pharmaceutics-16-00815-f003:**
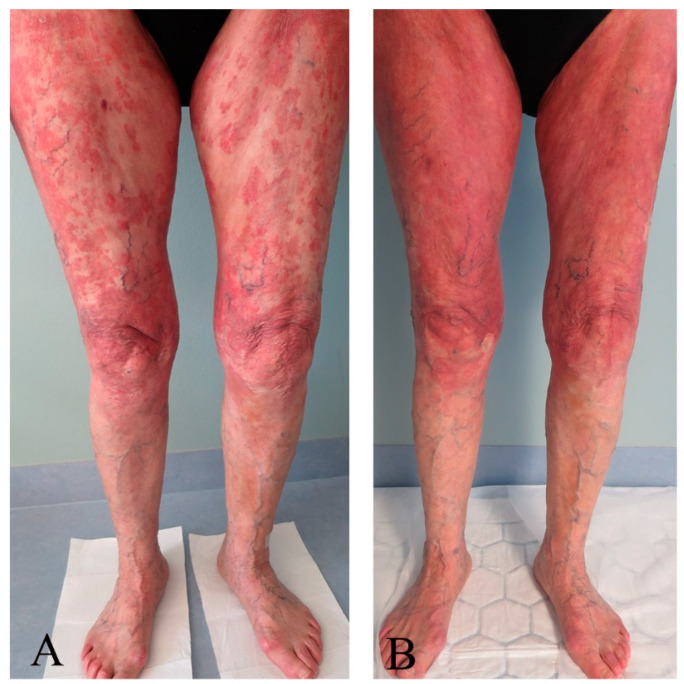
Images displaying the front of the lower extremities taken (**A**) before and (**B**) after 20 ALA-ECP treatments, with predominantly plaque localized to the thighs before treatment and with mainly macular erythema after treatment.

**Figure 4 pharmaceutics-16-00815-f004:**
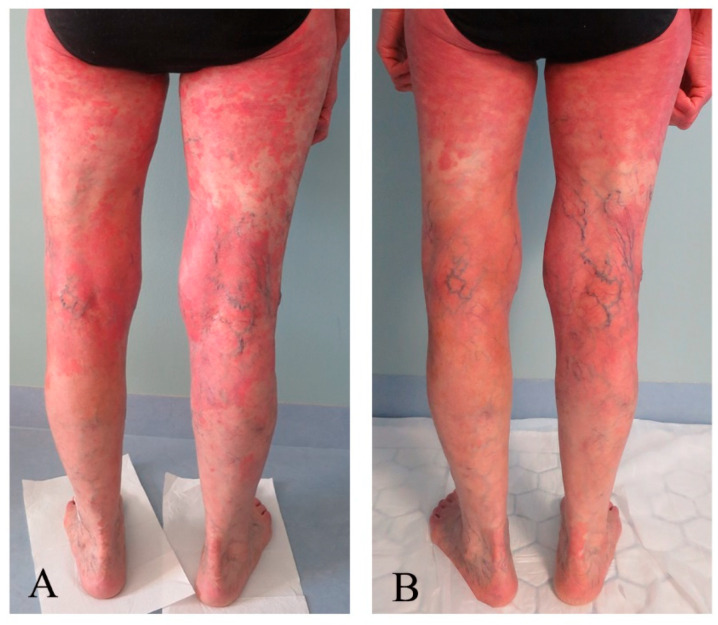
Images of the back of the lower extremities taken (**A**) before and (**B**) after 20 ALA-ECP treatments, with areas of erythema and plaque localized proximally and mid-extremity before and predominantly erythema proximally and with mild pigmentation mid-extremity, particularly on the left extremity after treatment.

**Table 1 pharmaceutics-16-00815-t001:** Measurements on blood pressure, heart rate, and temple temperature before and after each treatment and AST, ALT, bilirubin, and INR before treatment and after day 2 of each treatment cycle. All results are presented as Mean (min.–max.).

	Treatment Day 1		Treatment Day 2	
Test	Before	After	Before	After
Systolic BP, mm Hg	123 (99–159)	119 (99–142)	123 (93–145)	124 (102–144)
Diastolic BP, mm Hg	68 (53–86)	64 (52–89)	62 (47–80)	66 (52–80)
Pulse, beats/min	61 (51–73)	64 (59–71)	63 (54–73)	65 (55–76)
Temperature, °C	35.9 (35.6–36.2)	36.0 (35.8–36.3)	35.9 (35.5–36.2)	36.1 (36.0–36.3)
AST	30 (26–34)	n.a.	n.a	28 (25–38)
ALT	26 (24–32)	n.a	n.a	24 (20–27)
Bilirubin	11 (9–12)	n.a	n.a	9 (8–10)
PT-INR	1.0 (1.0–1.3)	n.a.	n.a.	1.2 (1.1–1.2)

**Table 2 pharmaceutics-16-00815-t002:** Laboratory analyses at baseline and before each treatment cycle.

Test [Reference Values]	Baseline	Mean (Min.–Max.)
Haemoglobin g/dL [11.7–15.3]	12.8	12.2 (11.4–13.1)
White blood cells 10^9^/L [39–73]	4.8	4.4 (3.7–5.6)
Diff. neutrophils % [39–73]	57	61 (51–75)
Diff. lymphocytes % [18–48]	24	19 (14–22)
Diff. monocytes % [5–13]	15	15 (7–21)
Diff. eosinophils % [0–8]	4	4 (1–5)
Diff. basophils % [0.2–1.3]	1.0	1.0 (1.0–1.0)
Platelet count 10^9^/L [164–370]	276	264 (220–280)
Mean cell volume fL [81–95]	94	91.7 (90.0–93.0)
Mean cell haemoglobin pg [27.1–32.6]	30.6	29.3 (28.5–30.3)
PT-INR [0.9–1.2]	1.3	1.0 (1.0–1.0)
Albumin g/L [36–45]	38	37.2 (34–40)
Protein, total g/L [62–78]	51	57 (51–62)
ALT U/L [10–45]	26	26.9 (24–32)
Alkaline phosphatase U/L [35–105]	79	70.8 (62–87)
AST U/L [15–35]	26	30.6 (26–34)
Bilirubin µmol/L [5–25]	12	10.2 (8–12)
Cholesterol mmol/L [3.9–7.8]	3.8	4.5 (4.0–5.2)
C-reactive protein mg/L [<5]	5	7 (5–24)
Glycated hemoglobin A1c mmol/mol [28–40]	35	35.6 (35–37)
Gamma-glutamyltransferase U/L [10–75]	35	24.6 (17–34)
Creatinine µmol/L [45–90]	46	46.8 (42–51)
Lactate dehydrogenase U/L [105–205]	212	219 (193–233)
Potassium mmol/L [3.6–4.6]	3.2	3.4 (3.3–3.6)
Sodium mmol/L [136–145]	141	141 (139–143)
Caicium cation mmol/L [2.15–2.51]	2.22	2.29 (2.20–2.35)
Erythrocyte sedimentation rate mm/h [≤17]	7	(7–24)
Lymphocyte subset CD4/CD8 ratio	15.9	8.6 (7.4–9.5)

**Table 3 pharmaceutics-16-00815-t003:** Adverse Events with frequency and assessment of the severity of the events and their relationship to study medication and Conceivable Adverse Events with frequency and severity of the events.

Type of Conceivable Event	Grade 1, No	Grade 2, No	Relation to Study Medication
Headache	2	0	Not applicable
Photosensitivity	0	1	Not applicable
Chills	2	1	Not applicable
**Type of adverse event**			
Soreness in skin	1	0	Possibly
Clotting in photo chamber	1	0	Possibly

no: number of events. **Grading of conceivable events.** Headache, Grade 1: Mild pain Photosensitivity, Grade 2: Tender erythema covering 10–30% body surface area. Chills, Grade 1: Mild sensation of cold; shivering; chatting of teeth; Grade 2: Moderate tremor of the entire body; narcotics indicated. **Grading of adverse events**. Grade 1 severity: Mild; asymptomatic or mild symptoms; clinical or diagnostic observation only; intervention not indicated.

**Table 4 pharmaceutics-16-00815-t004:** Skin and quality of life assessment scores at baseline and at the last control.

Target	Scoring Tool	Scoring Scale	Baseline	Last Control
Skin	mSWAT	0–100 (0 = no symptom)	93	29
Itch	Visual analog scale	0–10 (0 = no symptom)	8	4
Skindex, emotions	Questionnaire	0–100 (0 = no symptom)	48	3
Skindex, symptoms	Questionnaire	0–100 (0 = no symptom)	39	25
Skindex, function	Questionnaire	0–100 (0 = no symptom)	2	4
Skindex, single item	Questionnaire	0–100 (0 = no symptom)	50	0
EORTC30, functional	Questionnaire	0–100% (0 = low function)	100	100
EORTC30, symptoms	Questionnaire	0–100% (0 = low level of symptoms)	0	0
EORTC30, global health status	Questionnaire	0–100% (0 = low state)	75	100

## Data Availability

Data are contained within the article.

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
