# Peer review of "Application of Photodynamic Therapy with 5-Aminolevulinic Acid to Extracorporeal Photopheresis in the Treatment of Cutaneous T-Cell Lymphoma: A First-in-Human Phase I/II Studyâ€"

_pharmaceutics, 2024, doi:10.3390/pharmaceutics16060815_

Round 1
Reviewer 1 Report
Comments and Suggestions for Authors
In the present manuscript the authors report the Phase I /Phase II study of ALA in PDT treatment of cutaneous T-Cell lymphoma. The manuscript is well-written and flows smoothly. The introduction is solid and provides clear justification for the described study while describing the state of the art of ECP. The work is strictly related to the team’s previous study reported in ref 12 as clearly stated by the authors. The methodology is reported in detail and the results are critically analyzed and commented on in the manuscript. Moreover, the results after the treatment cycle seem to be very promising. The only arguable point is related to the single patient nature of the study which does not allow a statistical evaluation of the treatment. Nevertheless, at the present stage the works should be considered as an initial “proof of efficacy” for further clinical studies. There are just two minor points which need to be addressed in my opinion.
- The skin assessment shows an impressive improvement between the baseline and the last control. Were these improvements notable since the first treatment and increasing during the 20 treatments?
- How likely a second cycle of treatments could improve the results even further?
Author Response
Reviewer-1
In the present manuscript the authors report the Phase I /Phase II study of ALA in PDT treatment of cutaneous T-Cell lymphoma. The manuscript is well-written and flows smoothly. The introduction is solid and provides clear justification for the described study while describing the state of the art of ECP. The work is strictly related to the team’s previous study reported in ref 12 as clearly stated by the authors. The methodology is reported in detail and the results are critically analyzed and commented on in the manuscript. Moreover, the results after the treatment cycle seem to be very promising. The only arguable point is related to the single patient nature of the study which does not allow a statistical evaluation of the treatment. Nevertheless, at the present stage the works should be considered as an initial “proof of efficacy” for further clinical studies. There are just two minor points which need to be addressed in my opinion.
- The skin assessment shows an impressive improvement between the baseline and the last control. Were these improvements notable since the first treatment and increasing during the 20 treatments?
Reply: We assessed the treatment response every 3-month according to the approved study protocol, and saw a tendency of improvements with increasing number of treatment as shown in the table below. However, we do not have response data after the first or second treatments.
|
Test |
Screening (090419) |
3-month (240719) |
6-month (151019) |
9-month (070120) |
|
Skin assessment mSWAT, ( % BSA involved) |
93 (68) |
59 (51) |
53 (46) |
29 (27) |
- How likely a second cycle of treatments could improve the results even further?
Reply: See our above response for the point-1.
Reviewer 2 Report
Comments and Suggestions for Authors
The authors report a novel treatment procedure and chemical for photodynamic Cutaneous T-Cell Lymphoma treatment. The results look promising and positive, especially the improvement in skin quality and absence of liver toxicity. I only have minor comments.
11. Page 1 line 40 reference missing
22. Page 1 line 45 reference missing
33. Page 2 line 56 reference missing
44. Page 2 line 64 two different font sizes not consistent with the rest of the text
55. The utilization of cargoes for selectively delivering the ALA1 could allow for avoiding to remove blood from the patient treating it and donating it back. Such systems are currently in development and can be produced in an automated manner.2 This would be a promising outlook improving patient comfort.
References
(1) Kolesnikova, T. A.; Kiragosyan, G.; Le, T. H. N.; Springer, S.; Winterhalter, M. Protein A Functionalized Polyelectrolyte Microcapsules as a Universal Platform for Enhanced Targeting of Cell Surface Receptors. ACS Appl. Mater. Interfaces 2017, 9 (13), 11506–11517. https://doi.org/10.1021/acsami.7b01313.
(2) Li, W.; Gai, M.; Rutkowski, S.; He, W.; Meng, S.; Gorin, D.; Dai, L.; He, Q.; Frueh, J. An Automated Device for Layer-by-Layer Coating of Dispersed Superparamagnetic Nanoparticle Templates. Colloid J. 2018, 80 (6), 648–659. https://doi.org/10.1134/S1061933X18060078.
Here are my additional comments:
1. Methodology: Standard, no complaints. I would not know more robust methods, which could be applied ethically.
2. Results: are as good as they can be ethically be obtained ethically approvable. I consider the results to be interesting. And methods are explained in detail. One general problem is the fact, that only two patients were investigated. The authors could elaborate, if more patients are planned to be investigate, or if they have so few fitting patients, if the applicability of the method is not just severely limited.
3. The discussion should also focus in at least one additional paragraph on the case, if the method is expandable, and if more patients might be treated, or if in the whole area of Norway all cases are resolved and every country has just a few patients where this method can be applied at all.
4. The authors should put the conclusions into a own section.
Author Response
Reviewer-2
The authors report a novel treatment procedure and chemical for photodynamic Cutaneous T-Cell Lymphoma treatment. The results look promising and positive, especially the improvement in skin quality and absence of liver toxicity. I only have minor comments.
- Page 1 line 40 reference missing
Reply: Reference numbers are highlighted in the revised version.
- Page 1 line 45 reference missing
Reply: Reference number is highlighted in the revised version.
- Page 2 line 56 reference missing
Reply: Reference number was added in the revised version.
- Page 2 line 64 two different font sizes not consistent with the rest of the text
Reply: The font & size have been corrected in the revised version.
- The utilization of cargoes for selectively delivering the ALA1could allow for avoiding to remove blood from the patient treating it and donating it back. Such systems are currently in development and can be produced in an automated manner.2 This would be a promising outlook improving patient comfort.
References
(1) Kolesnikova, T. A.; Kiragosyan, G.; Le, T. H. N.; Springer, S.; Winterhalter, M. Protein A Functionalized Polyelectrolyte Microcapsules as a Universal Platform for Enhanced Targeting of Cell Surface Receptors. ACS Appl. Mater. Interfaces 2017, 9 (13), 11506–11517. https://doi.org/10.1021/acsami.7b01313.
(2) Li, W.; Gai, M.; Rutkowski, S.; He, W.; Meng, S.; Gorin, D.; Dai, L.; He, Q.; Frueh, J. An Automated Device for Layer-by-Layer Coating of Dispersed Superparamagnetic Nanoparticle Templates. Colloid J. 2018, 80 (6), 648–659. https://doi.org/10.1134/S1061933X18060078.
Here are my additional comments:
Reply: Although we are not sure if such cargoes are available today for a clinical trial, it is a very interesting suggestion and we will consider the possibility in our future studies.
1. Methodology: Standard, no complaints. I would not know more robust methods, which could be applied ethically.
Reply: Thanks.
Results: are as good as they can be ethically be obtained ethically approvable. I consider the results to be interesting. And methods are explained in detail. One general problem is the fact, that only two patients were investigated. The authors could elaborate, if more patients are planned to be investigate, or if they have so few fitting patients, if the applicability of the method is not just severely limited.
Reply: We fully agree with the comment. It was largely due to too few qualified patients available at our hospital.
The discussion should also focus in at least one additional paragraph on the case, if the method is expandable, and if more patients might be treated, or if in the whole area of Norway all cases are resolved and every country has just a few patients where this method can be applied at all.
Reply: The methodology of this new version of photopheresis is clinically feasible and patients well tolerated. This finding is not only based on the results of this study, but also on our previously published report using the same technique to treat the patients with cGvHD. We appreciate the suggestion to organize a national clinical trial with the possibility of including more relevant patients. In addition, we have made a paragraph to discuss possible mechanism of action of this technique that would provide biological basis for broadening applications of this technology to other diseases (Lines 304-311).
The authors should put the conclusions into a own section.
Reply: We have followed the comment and made a separate paragraph for ‘Conclusion’ in the revised version.
Reviewer 3 Report
Comments and Suggestions for Authors
The paper about the use of 5-aminolevulinic 2 Acid for Extracorporeal Photopheresis in the first-in-human phase I/II study is interesting, well written and in the field of a journal as Pharmaceutics. I have only one minor remarks.
In the “procedures” how is measured the “remaining lamp life”, or more clearly, how is measured the quantity UVA light received, which of course is a very important criterium. And does the wavelength amplitude remain constant over time? Is there any correlation between the intensity of the irradiation of the cells and some of the minor adverse effects observed only in some cases?
Author Response
Reviewer-3
The paper about the use of 5-aminolevulinic 2 Acid for Extracorporeal Photopheresis in the first-in-human phase I/II study is interesting, well written and in the field of a journal as Pharmaceutics. I have only one minor remarks.
In the “procedures” how is measured the “remaining lamp life”, or more clearly, how is measured the quantity UVA light received, which of course is a very important criterium. And does the wavelength amplitude remain constant over time? Is there any correlation between the intensity of the irradiation of the cells and some of the minor adverse effects observed only in some cases?
Reply: The built-in certified UV-A light source in the standard TherakosTM Photopheresis System with light mainly in the range of 320-410 nm was used. The intensity and homogeneity of the UV-A source emitting light were measured and controlled with an Ocean Optics spectrometer. It is expected to enhance cell killing with increased intensity of UV-A exposure, although this report has not studied this correlation.